# Effect of Subinhibitory Concentrations of Antibiotics and Disinfectants on IS*Aba*-Mediated Inactivation of Lipooligosaccharide Biosynthesis Genes in *Acinetobacter baumannii*

**DOI:** 10.3390/antibiotics10101259

**Published:** 2021-10-16

**Authors:** Héctor Olmeda-López, Andrés Corral-Lugo, Michael J. McConnell

**Affiliations:** Intrahospital Infections Laboratory, National Centre for Microbiology, Instituto de Salud Carlos III (ISCIII), 28221 Madrid, Spain; hectorolmeda97@gmail.com (H.O.-L.); acorral@isciii.es (A.C.-L.)

**Keywords:** *Acinetobacter baumannii*, insertion sequence, IS*Aba*, colistin, antibiotic resistance

## Abstract

Inactivation of the lipooligosaccharide (LOS) biosynthesis genes *lpxA*, *lpxC* and *lpxD* by IS*Aba* insertion elements results in high-level resistance to colistin in *A. baumannii*. In the present study, we quantify the rate of spontaneous insertional inactivation of LOS biosynthesis genes by IS*Aba* elements in the ATCC 19606-type strain and two multidrug clinical isolates. Using insertional inactivation of *lpxC* by IS*Aba*11 in the ATCC 19606 strain as a model, we determine the effect of several subinhibitory concentrations of the antibiotics, namely tetracycline, ciprofloxacin, meropenem, kanamycin and rifampicin, as well as the disinfectants ethanol and chlorhexidine on IS*Aba*11 insertion frequencies. Notably, subinhibitory concentrations of tetracycline significantly increased IS*Aba*11 insertion, and rifampicin completely inhibited the emergence of colistin resistance due to IS*Aba*11 inactivation of *lpxC*. Sequencing of IS*Aba*11 insertion sites within the *lpxC* gene demonstrated that insertions clustered between nucleotides 382 and 618 (58.3% of unique insertions detected), indicating that this may be a hotspot for IS*Aba*11 insertion. The alignment of insertion sites revealed a semi-conserved AT-rich consensus sequence upstream of the IS*Aba*11 insertion site, suggesting that IS*Aba*11 insertion sites may be sequence-dependent. This study explores previously uncharacterized aspects regarding the acquisition of colistin resistance through insertional activation in LOS biosynthesis genes in *A. baumannii*.

## 1. Introduction

Multidrug-resistant strains of *Acinetobacter baumannii* have complicated the clinical management of infections caused by this pathogen and are a significant cause of morbidity in the healthcare setting [1]. The peptide antibiotic colistin is one of the few antimicrobials that retains activity against many multidrug-resistant *A. baumannii* strains, although the emergence of pandrug-resistant isolates with decreased susceptibility to all clinically available antibiotics, including colistin, has been described [2,3]. The main bacterial target for colistin is the lipooligosaccharide (LOS) on the Gram-negative outer membrane, which interacts with positively charged colistin molecules due to its high negative charge [4]. This interaction results in membrane disruption and bacterial cell death in a wide range of Gram-negative species [4].

*A. baumannii* can acquire resistance to colistin via the modification of LOS through the addition of ethanolamine moieties to lipid A mediated by horizontally acquired Mcr-1 or mutations resulting in increased activity of endogenous PmrC [3]. Additionally, *A. baumannii* can acquire resistance to colistin (MIC values of ≥128 mg/L) through the complete loss of LOS biosynthesis due to a mutation in the genes that encode the first three steps of lipid A biosynthesis, *lpxA*, *lpxC* and *lpxD* [5,6]. Since LOS is the bacterial component that binds colistin through electrostatic interactions, the complete loss of LOS results in high-level resistance to colistin in A. baumannii [3,4,5]. Inactivation of these genes can occur through spontaneous point mutations, deletions and insertions, including the insertion of endogenous IS*Aba* insertion elements present in the *A. baumannii* genome [3]. IS*Abas* are genetic elements present in some *A. baumannii* strains that typically encode a transposase and are flanked by sequence repeats, facilitating their spontaneous insertion in the *A. baumannii* genome [7].

In the present study, we aimed to characterize the frequency of spontaneous emergence of colistin resistance in different *A. baumannii* strains through IS*Aba*-mediated insertional inactivation of LOS biosynthesis genes. In addition, we characterized the effect of subinhibitory concentrations of clinically relevant antibiotics and disinfectants on the frequency of IS*Aba* insertion and identified a semi-conserved sequence motif at IS*Aba* insertion sites.

## 2. Results

### 2.1. Development of an Assay for Quantifying ISAba Insertion Frequency in LOS Biosynthesis Genes

In order to characterize the spontaneous emergence of colistin resistance via IS*Aba*-mediated insertional inactivation of LOS biosynthesis genes under different conditions, a PCR-based assay was developed that permitted the identification and quantification of *lpxA*, *lpxC* and *lpxD* mutants harboring IS*Aba* elements. In this assay, *A. baumannii* strains are cultured under defined conditions (e.g., in the presence or absence of an antibiotic; see Materials and Methods) for 18 h and then plated on media containing 10 mg/L of colistin or media without antibiotic. Colistin-resistant derivatives that grow in the presence of 10 mg/L of colistin are isolated and subjected to PCR with primers flanking the *lpxA*, *lpxC* and *lpxD* genes (Appendix A) in order to identify genes harboring the characteristic insertion of approximately 1.1 Kb that corresponds to IS*Aba* elements (Figure 1a). The presence of IS*Aba* elements in these amplicons is confirmed by DNA sequencing. The quantification of colistin-resistant derivatives containing IS*Aba* insertions and total bacteria in the culture permits calculation of the frequency of spontaneous IS*Aba* insertion in each gene under the conditions tested, and is expressed as the number of isolates harboring an IS*Aba* element/10^9^ colony-forming units (CFU).

### 2.2. Characterization of ISAba Insertion Frequencies in LOS Biosynthesis Genes in Different A. baumannii Strains

Using the assay described above, we first aimed to characterize the baseline frequency of IS*Aba* insertion in *lpxA*, *lpxC* and *lpxD* for the ATCC 19606-type strain, which harbors IS*Aba*11 [6], and two multidrug-resistant clinical isolates that have previously been shown to harbor IS*Aba* elements, Ab-167 and Ab-176 [8,9]. All strains were grown in the absence of antibiotics, and IS*Aba* insertion frequencies were quantified as described above in three independent biological replicates. As can be seen in Figure 1b, insertion frequencies for the ATCC 19606 strain were 1.85 ± 0.93, 8.34 ± 2.25 and 0.98 ± 0.98 /10^9^ CFU for *lpxA*, *lpxC* and *lpxD*, respectively. For the multidrug-resistant clinical isolate Ab-167, there was a tendency towards higher insertion frequencies in all genes compared to the ATCC 19606 strain, with frequencies of 19.02 ± 9.06, 86.41 ± 57.04 and 9.52 ± 5.79 /10^9^ CFU for *lpxA*, *lpxC* and *lpxD*, respectively. Finally, the Ab-176 strain demonstrated a tendency towards the lowest insertion frequencies of the three strains tested with no insertions detected in *lpxA*, 1.41 ± 1.41 /10^9^ CFU in *lpxC* and 0.61 ± 0.61 /10^9^ CFU in *lpxD*. Interestingly, there was a tendency towards higher insertion frequencies in *lpxC* compared to *lpxA* and *lpxD* for all three strains tested.

### 2.3. Effect of Antibiotics and Disinfectants on ISAba11 Insertion Frequencies in lpxC

We next aimed to determine how subinhibitory concentrations of clinically relevant antibiotics and disinfectants affected IS*Aba* insertion frequencies in LOS biosynthesis genes. Based on the finding that insertion frequencies in *lpxC* tended to be higher than for *lpxA* and *lpxD* (Figure 1b), we used IS*Aba* insertion in *lpxC* in the ATCC 19606 strain for these studies. The ATCC 19606 strain harbors IS*Aba*11 in its genome [6], and its susceptibility profile permits the use of clinically relevant concentrations of multiple antimicrobials. The minimum inhibitory concentrations of five antibiotics (tetracycline, ciprofloxacin, meropenem, kanamycin and rifampicin) and two disinfectants (ethanol and chlorhexidine) where determined for the ATCC 19606 strain by broth microdilution. Subsequently, the effect of subinhibitory concentrations of these compounds was quantified by incubating cultures of the ATCC 19606 strain with one-quarter of the MIC of each compound and determining IS*Aba*11 insertion frequencies in *lpxC* using the assay described above. As can be seen in Figure 2a, the frequency of IS*Aba*11 insertion in *lpxC* (grey bars) and the frequency of emergence of colistin resistance (white bars) was similar to control samples incubated without an antibiotic for ciprofloxacin, meropenem, kanamycin, ethanol and chlorhexidine. In contrast, incubation with tetracycline resulted in significantly increased IS*Aba*11 insertion in *lpxC* (*p* = 0.04), whereas incubation with rifampicin resulted in the complete absence of isolates with IS*Aba*11 insertions in *lpxC* (*p* = 0.02). The effect of subinhibitory concentrations of rifampicin on IS*Aba*11 insertion in *lpxC* was further explored by characterizing the effect of 1/8 to 1/2 of the MIC of rifampicin. As can be seen in Figure 2b, all concentrations of rifampicin tested resulted in the complete absence of isolates with IS*Aba*11 insertions in *lpxC*.

### 2.4. Analysis of ISAba11 Insertion Sites in lpxC

In order to characterize IS*Aba*11 insertion sites in *lpxC*, amplicons containing IS*Aba*11 in *lpxC,* obtained via PCR with primers flanking the *lpxC* gene as described in Appendix A, were subjected to DNA sequencing. Amplicons from a total of 32 derivatives of the ATCC 19606 strain containing IS*Aba*11 insertions in *lpxC* were sequenced, resulting in the identification of 12 unique insertion sites within the *lpxC* gene (Figure 3). Interestingly, 7 of the 12 (58.3%) unique insertion sites were between nucleotides 382 and 618, with the other 5 falling outside this region. Alignment of the 12 unique insertion sites demonstrated conserved nucleotides in sequences flanking the insertion site. At the −3 position with respect to the insertion site, 8 of 12 sequences contained an A; at the −8 position, 8 of 12 contained an A; and at the −11 position, 8 of 12 contained a T. These data suggest semi-conserved sequence motifs may facilitate IS*Aba*11 insertion.

## 3. Discussion

Insertion sequences are known to contribute to genome plasticity in Gram-negative bacteria, including A*. baumannii* [10,11]. Based on the IS finder database (https://www-is.biotoul.fr) (accessed on: 10 October 2021), fifty-five IS sequences have been identified in *A. baumannii* to date [12]. IS*Abas* have been shown to contribute to antibiotic resistance through multiple mechanisms, including insertional inactivation of genes encoding antibiotic targets and increasing the expression of antibiotic resistance genes, such as OXA-58 and efflux pumps, through insertion in regions upstream of genes [13]. However, in spite of the importance of IS*Abas* in contributing to antimicrobial resistance, quantitative characterization of insertion frequencies and their strain dependence have not been fully characterized. In this study, we developed a novel assay that uses colistin resistance produced by insertional inactivation of the lipooligosaccharide biosynthesis genes *lpxA*, *lpxC* and *lpxD* to assess the frequency of IS*Aba*-mediated gene inactivation in *A. baumannii*. This assay allowed us to characterize IS*Aba* insertion frequencies in these genes in different *A. baumannii* strains (Figure 1b). The results of this assay may suggest intrinsic differences between strains with respect to IS*Aba* insertion frequencies as insertion frequencies in all three genes tested were approximately 10-fold higher in the Ab-167 strain compared to the Ab-176 strain. These results may also suggest differences in IS*Aba* insertion between different genes within a strain, as insertion frequencies in the *lpxC* gene were consistently higher for all strains tested. The mechanisms underlying these potential differences have not been elucidated.

Our results also indicate that subinhibitory concentrations of different antibiotics can affect IS*Aba* insertion frequency. As shown in Figure 2, exposure to tetracycline, which inhibits protein synthesis via binding to the 30 S ribosomal subunit, significantly increased the IS*Aba* insertion frequency in *lpxC* compared to untreated controls. The mechanism underlying this increased frequency was not assessed here; however, it is interesting to note that antimicrobials have previously been shown to promote the mobility of transposable elements through the activation of the SOS pathway [14], and a previous study demonstrated that tetracycline could induce SOS-mediated acquisition of antibiotic resistance in *A. baumannii* [15]. In contrast to the results seen with tetracycline, the presence of subinhibitory concentrations of rifampicin, which inhibits bacterial RNA synthesis, completely inhibited the emergence of mutants with IS*Aba* insertions in *lpxC* at all concentrations tested (Figure 2b). We previously demonstrated that *A. baumannii* strains acquiring resistance to colistin via insertional inactivation of LOS biosynthesis genes become highly susceptible to rifampicin, with MIC values ≤ 0.125 mg/L [16]. These previous results may suggest that spontaneous mutants with IS*Aba*-mediated inactivation of LOS biosynthesis genes are not viable in the presence of the concentrations of rifampicin used in this study, thus explaining the complete absence of these mutants observed here (Figure 2).

The isolation of a large number of independent mutants with IS*Aba* insertions in *lpxC* allowed us to sequence multiple IS*Aba* insertion sites and assess insertion sites within *lpxC*. Our findings that IS*Aba* insertions clustered between nucleotides 382 and 618 of *lpxC* are consistent with a previous study suggesting two possible insertional hot spots within *lpxC* between nucleotides 390 and 393 and nucleotides 420 and 421 [6]. Sequence analysis of insertion sites also showed a bias towards A at the −3 and −8 positions and T at the −11 position. These results may suggest that IS*Aba* insertion is sequence-biased in *A. baumannii*.

Taken together, the results of this study elucidate previously uncharacterized aspects related to the emergence of resistance to a last-resort antibiotic in a clinically important pathogen due to the insertional activation of genes by mobile genetic elements.

## 4. Materials and Methods

### 4.1. Strains and Growth Conditions

The colistin-susceptible *A. baumannii* ATCC 19606 strain and two previously described multidrug-resistant (colistin susceptible) clinical isolates, Ab-167 and Ab-176, which were previously shown to harbor IS*Aba* elements, were included in this study [8,9]. *A. baumannii* strains were routinely grown at 37 °C and maintained in Mueller–Hinton Broth II (MHBII) or Luria Bertani media (LB), unless otherwise stated.

### 4.2. Antibiotics, Disinfectants and Minimum Inhibitory Concentration Determination

Colistin sulfate (COL) and ethanol (EtOH) were purchased from PanReac AppliChem ITW Reagents. Ciprofloxacin (CIP), chlorhexidine (CLX), kanamycin sulfate (KAN), meropenem trihydrate (MEM), rifampicin (RIF) and tetracycline hydrochloride (TET) were purchased from Sigma-Aldrich. All antimicrobials were prepared in sterile milliQ water at the desired concentration. MIC values were determined for disinfectants and antibiotics. Overnight cultures of each strain were adjusted in MHBII to 5 × 10^5^ CFU/mL and MIC values were determined according to the recommendations of the Clinical and Laboratory Standards Institute guidelines [17].

### 4.3. Quantification of ISAba Insertion in LOS Biosynthesis Genes

For determining IS*Aba* insertion frequencies, frozen stocks of *A. baumannii* strains were plated on LB media and incubated overnight at 37 °C, and colonies were used to inoculate a pre-culture of LB broth before incubation at 37 °C for 18 h. Pre-cultures were adjusted to an OD_600_ of 2, and a 1:100 dilution was prepared in LB broth (with or without antibiotics, as indicated) before incubation for 18 h at 37 °C with agitation at 200 rpm. Spontaneous colistin-resistant mutants were identified by plating 100 μL of the culture on MH agar plates containing 10 mg/L of colistin followed by incubation overnight at 37 °C. Colistin-resistant colonies were quantified and individually struck onto MH plates containing 10 mg/L of colistin for the detection of IS*Aba* insertion by PCR. Cultures were also quantitatively plated on LB agar with no antibiotics in order to determine the number of colony-forming units (CFU)/mL in the culture.

For the detection of IS*Aba* insertions in colistin-resistant isolates, genomic DNA was obtained from the colistin-resistant mutants by incubating a cellular pellet in 20 µL of water at 98 °C for 10 min, followed by centrifugation at 13,000 rpm for two minutes. Supernatants were used to amplify the LOS biosynthesis genes *lpxA, lpxC* and *lpxD* using the primers described by Moffat et al. [5] (Appendix A). PCR products were separated on a 1% agarose gel to identify mutants with IS*Aba* insertions. Amplicons demonstrating the expected size corresponding to IS*Aba* insertions were sequenced in order to confirm the presence of an IS*Aba*. The IS*Aba* insertion rate was calculated using Equation (1).
(1)$$\mathrm{IS}Aba\;\mathrm{Insertion}\;\mathrm{frequency}=\frac{\mathrm{CFU}/\mathrm{mL}\;\mathrm{with}\;\mathrm{IS}Aba\;\mathrm{insertion}}{\mathrm{Total}\;\mathrm{CFU}/\mathrm{mL}}\times 10^{9}$$

For assays assessing the effect of antibiotics and disinfectants, the following concentrations were employed: COL, 10 mg/L; TET, 0.5 mg/L; CIP, 1 mg/L; MEM, 0.5 mg/L; KAN, 2 mg/L; RIF, 0.5 mg/L; EtOH, 1.5%; and CLX, 8 mg/L.

### 4.4. Sequencing of ISAba11 Insertions in lpxC

Amplicons containing an IS*Aba*11 insertion in the *lpxC* gene from assays employing the ATCC 19606 strain were subjected to sequencing with the primers IS*Aba*11_Fw_Seq and IS*Aba*11_Rev_Seq (Appendix A), and the insertion site was determined using the BioEdit sequence alignment editor.

### 4.5. Statistical Analysis

The distribution of all datasets was assessed using the Shapiro-Wilk test. IS*Aba* insertion and colistin resistance frequencies between strains were compared using a one-way ANOVA, and differences between groups were determined using the Tukey post hoc test. A Mann–Whitney U-test was performed to assess differences in the frequencies of the emergence of colistin-resistant mutants harboring IS*Aba*11 insertions and of the colistin-resistance emergence. A *p*-value ≤ 0.05 was considered significant.

## Figures and Tables

**Figure 1 antibiotics-10-01259-f001:**
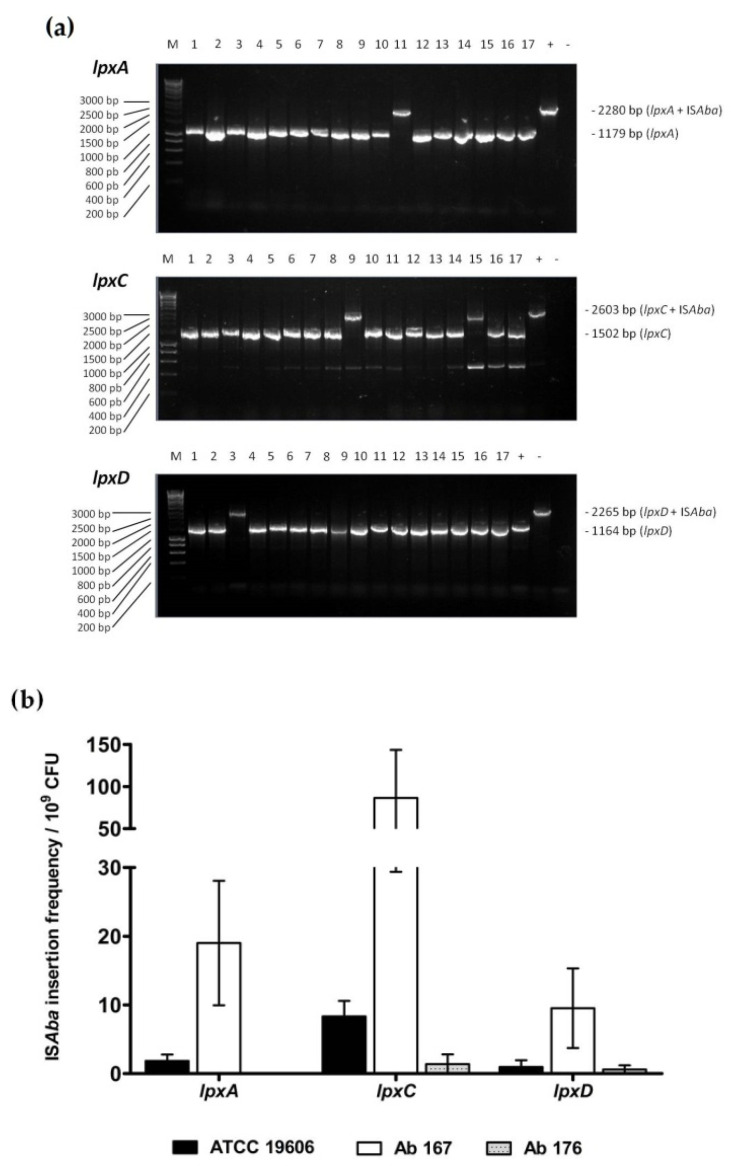
IS*Aba* insertion frequencies in *lpxA*, *lpxC* and *lpxD* genes in three *A. baumannii* strains. (**a**) Representative agarose gels showing PCR products obtained after amplifying *lpxA*, *lpxC*, and *lpxD* genes in order to screen colistin-resistant *A. baumannii* mutants for the presence of IS*Aba* insertions using the newly developed method described in the Materials and Methods section. (**b**) IS*Aba* insertion rates in *lpxA*, *lpxC* and *lpxD* genes in the *A. baumannii* strains ATCC 19606, Ab-167 and Ab-176 determined using the method described in the Materials and Methods section. M, DNA marker; +, positive control with a sequence-confirmed IS*Aba*11 insertion; −, negative control; bp, base pairs.

**Figure 2 antibiotics-10-01259-f002:**
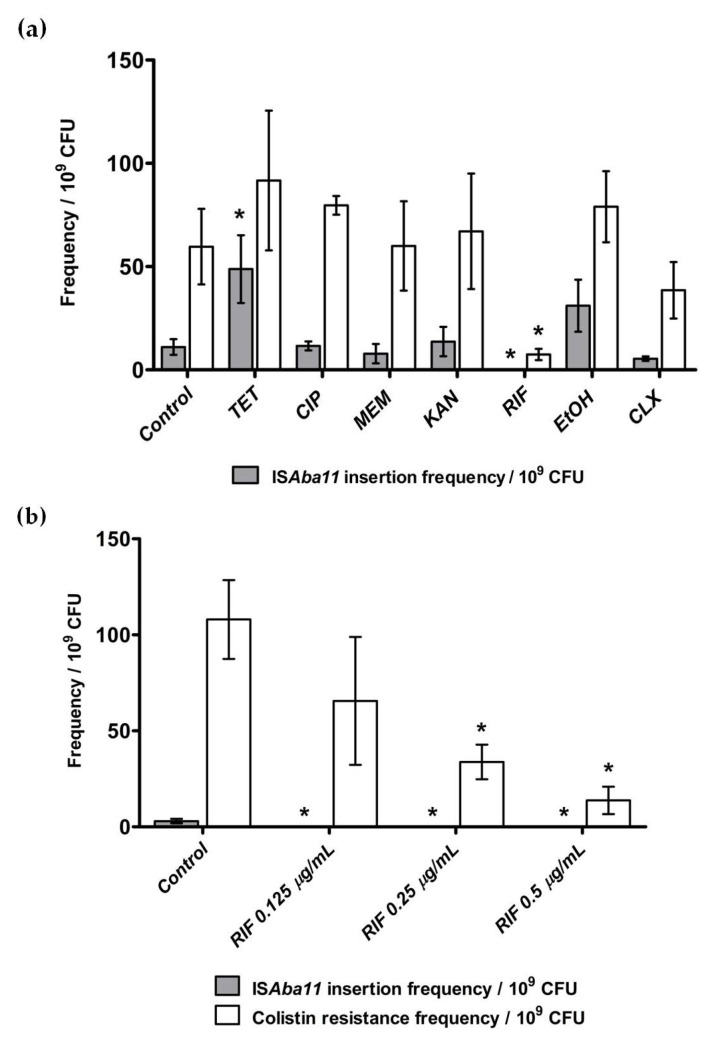
Effect of subinhibitory concentrations on IS*Aba*11 insertion frequency in *lpxC*. (**a**) Frequency of spontaneous emergence of colistin resistance (by any mechanism; white bars) and the emergence of colistin-resistant mutants harboring IS*Aba*11 insertions in *lpxC* (grey bars) after exposure to subinhibitory concentrations of different antibiotics and disinfectants at concentrations of one-quarter of the MIC; TET, tetracycline (0.5 µg/mL); CIP, ciprofloxacin (1 µg/mL); MEM, meropenem (0.5 µg/mL); KAN, kanamycin (2 µg/mL); RIF, rifampicin (0.5 µg/mL); EtOH, ethanol (1.5%) and CLX, chlorhexidine (8 mg/mL). (**b**) Spontaneous emergence of colistin resistance (by any mechanism; white bars) and frequency of colistin-resistant mutants harboring IS*Aba*11 insertions in *lpxC* (grey bars) after exposure to the indicated concentrations of rifampicin. Data represented as mean ± standard error of the mean (SEM) of three independent experiments. (*, *p* < 0.05 compared to untreated controls).

**Figure 3 antibiotics-10-01259-f003:**
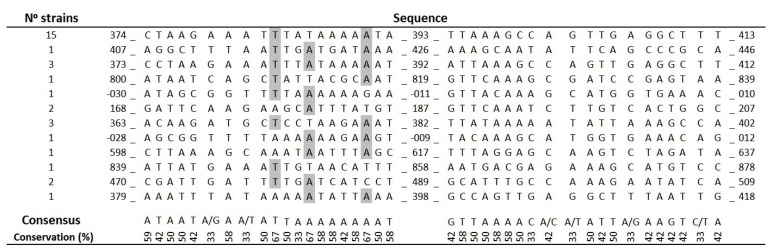
Molecular characterization of the IS*Aba*11 insertion in the *lpxC* gene. Alignment of IS*Aba*11 insertion sites in *lpxC* from independent colistin-resistant mutants. Numbers indicate the nucleotide sequence of the *lpxC* gene. Grey highlights indicate nucleotides that are more than 65% conserved between independent insertion sites.

## Data Availability

The data presented in this study are available on request from the corresponding author.

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
