# Peer review of "Effect of Subinhibitory Concentrations of Antibiotics and Disinfectants on ISAba-Mediated Inactivation of Lipooligosaccharide Biosynthesis Genes in Acinetobacter baumannii"

_antibiotics, 2021, doi:10.3390/antibiotics10101259_

Round 1

Reviewer 1 Report

Integration of inherent ISAba insertion elements into lipooligosaccharide (LOS) biosynthesis genes lpxA, lpxC and lpxD can lead to inactivation of LOS genes and can result in increased resistance to colistin in A. baumannii. In the present manuscript, the Authors have measured the frequency of inactivation of LOS genes by ISAba11 in different antibiotics in the ATCC 19606 type strain and two multidrug clinical isolates (Ab-167, and Ab-176).  Treatment with tetracycline resulted in higher ISAba11 insertion and colistin resistance while rifampicin treatment resulted in lower insertion and resistance. Overall, the study is interesting, but I have the following concerns and suggestions.

  1. In the introduction, please explain the mechanism behind the inactivation of LOS genes and the resulting increase in antibiotic resistance.
  2. Label the ISAba11 and ISAba11+LOS bands in figure 1A.
  3. It would be great to perform experiments described in Figure 2A using strains Ab-167, and Ab-176. This will help in understanding how resistant strains respond to different antibiotics.
  4. Please try to interpret the results presented in Figure 2A&B based on the mechanism of action of antibiotics.

Author Response

1. In the introduction, please explain the mechanism behind the inactivation of LOS genes and the resulting increase in antibiotic resistance.

RESPONSE: As suggested by the reviewer, we have included additional text in the introduction explaining how mutations in LOS biosynthesis genes lead to resistance to colistin.

2. Label the ISAba11 and ISAba11+LOS bands in figure 1A.

RESPONSE: We have labeled the bands in Figure 1A, as suggested by the reviewer.

3. It would be great to perform experiments described in Figure 2A using strains Ab-167, and Ab-176. This will help in understanding how resistant strains respond to different antibiotics.

RESPONSE: We agree with the reviewer that it would be interesting to characterize how ISAba insertion frequencies are affected by antibiotics in the clinical isolates Ab-167 and Ab-176. Unfortunately, these clinical isolates show high level resistance to all of the antibiotics used in the study.  Thus, it is not possible to perform assays with subinhibitory concentrations of antibiotics at concentrations that are clinically relevant.

4. Please try to interpret the results presented in Figure 2A&B based on the mechanism of action of antibiotics.

RESPONSE: We agree with the reviewer that this is an interesting point. We have included additional text in the discussion section placing the results presented in Figure 2 in the context of the different mechanisms of action of the antibiotics tested.

Reviewer 2 Report

The manuscript describes the characterization of the frequency of spontaneous emergence of colistin resistance in different A. baumannii strains through ISAba-mediated insertional inactivation of LOS biosynthesis genes. They also characterize the effect of subinhibitory concentrations of clinically-relevant antibiotics and disinfectants on the frequency of ISAba insertion.

Methods are adequate and the results are sound.

The results clearly show that subinhibitory concentrations of tetracycline significantly increased ISAba11 insertion and that rifampicin completely inhibited the emergence colistin resistance due to ISAba11 inactivation of  lpxC

By sequencing of the  identified amplicons containing ISAba11 153 in lpxC, a putative semi-conserved sequence motif at ISAba11 insertion sites is suggested.

Only minor concerns should be addressed.

Some difference in numbering of the sequenced nucleotides must be corrected and clarified.. 

these insertions were clustered between nucleotides 382 and 618, line 158, but numbering of insertions in figure 3 ranges from -030 to 878

Is there any difference in colistin resistance between both Ab-167 and Ab-176 multigrug resistant strains, that could explain the different behaviour concerning the ISAba insertions?

Author Response

Only minor concerns should be addressed.

1. Some difference in numbering of the sequenced nucleotides must be corrected and clarified..these insertions were clustered between nucleotides 382 and 618,line 158, but numbering of insertions in figure 3 ranges from -030 to 878.

RESPONSE: We agree with the reviewer that this point was not sufficiently clear in the manuscript. We have modified the results section and the legend for Figure 3 in order to make it clear that the majority, but not all (58.3%), insertions were found between nucleotides 382 and 618.

2. Is there any difference in colistin resistance between both Ab-167 and Ab-176 multidrug resistant strains, that could explain the different behaviour concerning the ISAba insertions?

RESPONSE: The reviewer raises an interesting point that we had not mentioned in the manuscript.  All strains included in the study (Ab-167, Ab-176 and ATCC 19606) are susceptible to colistin with MIC values between 0.5-1.0 mg/L, suggesting that differences in susceptibility to colistin do not account for the observed difference in ISAba insertion frequencies. We have indicated in the methods section that all strains are susceptible to colistin in order to make this point clear.